# Deep learning-based electrocardiographic screening for chronic kidney disease

Lauri Holmstrom[1,2,3,4,10], Matthew Christensen[1,4,10], Neal Yuan[5], J. Weston Hughes[6], John Theurer[1,4], Melvin Jujjavarapu[7], Pedram Fatehi[8], Alan Kwan [1], Roopinder K. Sandhu[1], Joseph Ebinger[1], Susan Cheng[1], James Zou [6,9], Sumeet S. Chugh[1,2,4,11] & David Ouyang [1,4,11 ✉]

### Abstract

**Background** Undiagnosed chronic kidney disease (CKD) is a common and usually asymptomatic disorder that causes a high burden of morbidity and early mortality worldwide. We developed a deep learning model for CKD screening from routinely acquired ECGs.

**Methods** We collected data from a primary cohort with 111,370 patients which had 247,655 ECGs between 2005 and 2019. Using this data, we developed, trained, validated, and tested a deep learning model to predict whether an ECG was taken within one year of the patient receiving a CKD diagnosis. The model was additionally validated using an external cohort from another healthcare system which had 312,145 patients with 896,620 ECGs between 2005 and 2018.

**Results** Using 12-lead ECG waveforms, our deep learning algorithm achieves discrimination for CKD of any stage with an AUC of 0.767 (95% CI 0.760–0.773) in a held-out test set and an AUC of 0.709 (0.708–0.710) in the external cohort. Our 12-lead ECG-based model performance is consistent across the severity of CKD, with an AUC of 0.753 (0.735–0.770) for mild CKD, AUC of 0.759 (0.750–0.767) for moderate-severe CKD, and an AUC of 0.783 (0.773–0.793) for ESRD. In patients under 60 years old, our model achieves high performance in detecting any stage CKD with both 12-lead (AUC 0.843 [0.836–0.852]) and 1-lead ECG waveform (0.824 [0.815–0.832]).

**Conclusions** Our deep learning algorithm is able to detect CKD using ECG waveforms, with stronger performance in younger patients and more severe CKD stages. This ECG algorithm has the potential to augment screening for CKD.

### Plain language summary

Chronic kidney disease (CKD) is a common condition involving loss of kidney function over time and results in a substantial number of deaths. However, CKD often has no symptoms during its early stages. To detect CKD earlier, we developed a computational approach for CKD screening using routinely acquired electrocardiograms (ECGs), a cheap, rapid, non-invasive, and commonly obtained test of the heart's electrical activity. Our model achieved good accuracy in identifying any stage of CKD, with especially high accuracy in younger patients and more severe stages of CKD. Given the high global burden of undiagnosed CKD, novel and accessible CKD screening strategies have the potential to help prevent disease progression and reduce premature deaths related to CKD.

[1] Department of Cardiology, Smidt Heart Institute, Cedars-Sinai Medical Center, Los Angeles, CA, USA. [2] Center for Cardiac Arrest Prevention, Department of Cardiology, Smidt Heart Institute, Cedars-Sinai Medical Center, Los Angeles, CA, USA. [3] Research Unit of Internal Medicine, Medical Research Center Oulu, University of Oulu and Oulu University Hospital, Oulu, Finland. [4] Division of Artificial Intelligence in Medicine, Department of Medicine, Cedars-Sinai Medical Center, Los Angeles, CA, USA. [5] Department of Medicine, Division of Cardiology, San Francisco VA, UCSF, San Francisco, CA, USA. [6] Department of Computer Science, Stanford University, Palo Alto, CA, USA. [7] Enterprise Information Service, Cedars-Sinai Medical Center, Los Angeles, CA, USA. [8] Division of Nephrology, Department of Medicine, Stanford University, Palo Alto, CA, USA. [9] Department of Biomedical Data Science, Stanford University, Palo Alto, CA, USA. [10] These authors contributed equally: Lauri Holmstrom, Matthew Christensen. [11] These authors jointly supervised this work: Sumeet S. Chugh, David Ouyang. ✉email: David.Ouyang@cshs.org

Almost 700 million individuals globally have chronic kidney disease (CKD), an important but often unrecognized cause of morbidity and early mortality[1]. The initial presentation of CKD is usually asymptomatic and without overt clinical manifestations especially in the early stages of the disease. Recently, the Global Burden of Diseases, Injuries and Risk Factors Study (GBD) estimated that CKD accounts for 4.6% of total mortality worldwide, with a 41.5% increase between 1990 and 2017[1]. Delayed diagnosis and limited patient recognition of the condition contribute significantly to the burden of morbidity[2,3]. Early detection can potentially change the disease trajectory. The most common causes of CKD, such as hypertension and diabetes, can be reversible or treatable, and early diagnosis is crucial for avoiding renal replacement therapy[4,5]. There are few methods to cheaply or non-invasively screen for CKD, with conventional risk calculators lacking specificity and requiring both serum and urine laboratory testing[6].

Electrocardiograms (ECGs) are inexpensive, non-invasive, widely available, and rapid diagnostic tests frequently obtained during routine visits, prior to exercise, during preoperative evaluation, and for patients at increased risk of cardiovascular disease. Deep learning algorithms (DLA) have recently been applied to medical imaging and clinical data to achieve high precision, and to identify additional information beyond the interpretation of human experts[7,8]. Deep learning analysis of ECG waveforms has had potentially promising performance in prognosticating outcomes[9], identifying subclinical disease[10,11], and identifying systemic phenotypes not traditionally associated with ECGs[12,13]. Given the prior success in identifying occult arrhythmias[14,15], ventricular dysfunction[10], anemia[13], and age[12], DLA applied to screening ECGs could potentially identify patients who would benefit from further evaluation for kidney disease.

The high prevalence of concomitant cardiovascular disease and the well-established changes that accompany electrolyte abnormalities suggest that the ECG is also altered in the setting of CKD and that discrete electrocardiographic signatures could be identifiable with deep learning techniques. Patients with CKD have a disproportionate accumulation of cardiovascular risk factors, such as diabetes and hypertension, as well as subclinical cardiovascular changes such as left ventricular hypertrophy, myocardial fibrosis, and diastolic dysfunction[16]. It is not fully clear at which stage CKD patients start to develop manifest cardiovascular changes. However, recent studies have reported that in addition to coronary artery disease and left ventricle hypertrophy, patients with early-stage CKD may already have an increase in diffuse myocardial fibrosis on cardiac MRI as well[17]. It is hence likely that already early-stage CKD associates with non-specific ECG signals. In addition to myocardial remodeling, CKD associates with a variety of electrolyte abnormalities that also cause widespread ECG abnormalities (e.g., decreased T-wave amplitudes in hypokalemia, large-amplitude T-waves, and prolonged QRS duration in hyperkalemia, and QTc prolongation in hypocalcemia)[18]. Prior work has shown such patterns are detectable on ECG waveforms, contributing to the AI-ECG detection of hyperkalemia, which might augment a model's ability to detect CKD[19,20]. However, given the relative infrequency of overt abnormalities, likely not the primary feature analyzed in detecting CKD. Given such observations, it may be possible that asymptomatic CKD presents with subtle ECG alterations that are not visible to the human eye.

To overcome current limitations in screening for occult CKD, we designed, trained, and validated a deep learning model to predict CKD, including end-stage renal disease (ESRD), by analysis of waveform signals from a single 12-lead and 1-lead ECG. Incorporating both structured information from medical diagnoses as well as laboratory data, we assessed the ability of our model to evaluate the entire spectrum of kidney disease. To further evaluate our model, we validated its performance using corresponding data from a separate healthcare system.

## Methods

**Data sources and study population.** We retrospectively identified 64,308 ECGs among 7816 patients between 2005 and 2019 which were linked to a diagnosis of CKD within a 1-year window at Cedars-Sinai Medical Center. We also identified 183,290 ECGs among 103,554 patients between 2008 and 2019 with no CKD diagnoses at any point, which were used as matched negative controls. Study cohorts included both ambulatory and in-hospital patients. If a patient had multiple ECGs taken within a year of a CKD diagnosis, each ECG-CKD pair was considered an independent case during model training, but only one was used in the test datasets. The study population from Cedars-Sinai Medical Center was randomly split 8:1:1 into training, validation, and test cohorts by patient such that the multiple ECGs from the same patient were limited to one cohort. In addition, we identified 896,620 ECGs among 312,145 patients at Stanford Healthcare from 8/2005 to 6/2018, which were used for external validation (Fig. 1).

ECGs from Cedars-Sinai Medical Center were obtained from MUSE Cardiology Information System (GE Healthcare), and the model used the original ECG waveforms stored for training the model in CKD prediction. In external validation at Stanford University, ECGs were stored using the Phillips TraceMaster system, and were independently used as input examples for external validation. The ECG waveform data were acquired at a sampling rate of 500 Hz and extracted as 10 second, $12 \times 5000$ matrices of amplitude values. ECGs with missing leads were excluded from the study cohort. Associated clinical data for each patient was obtained from the electronic health record. The data on medical diagnoses was extracted from the electronic health records using International Classification of Diseases (ICD) 9/10th edition codes, which are listed in Supplementary Table 1. Demographic and clinical characteristics (e.g., age, gender, BMI, cardiovascular disease) were also extracted from the electronic health records. The institutional review boards of Cedars-Sinai Medical Center and Stanford Healthcare approved the study protocol (Cedars Protocol 1506 and Stanford Protocol 43721). Informed consent was waived for analysis of de-identified retrospective data.

**AI model design and training.** We designed a convolutional neural network, for ECG interpretation with potential for clinical data integration to predict the primary outcomes of chronic kidney disease and end-stage renal disease (Fig. 2). The model was trained to predict outcomes with the input of one 12-lead ECG obtained within 1 year of diagnosis. Please see Supplementary information for additional details on model training. If the same patient had multiple ECGs, each was considered an independent case. Models were trained using the PyTorch deep learning framework. The model was initialized with random weights and trained using a binary cross-entropy loss function for up to 100 epochs with an ADAM optimizer and an initial learning rate of 1e-4. Early stopping was performed based on the validation dataset's area under the receiver operating curve. Local Interpretable Model-agnostic Explanations (LIME)[15,21] was used with 1000 samples per study to identify relevant features in the ECG waveform by iteratively randomly perturbing 0.5% of the waveform and identifying which changes most impacted model performance.

**Statistical analysis.** All analyses were performed on the held-out test dataset, which was never seen during model training. The performance of the model in predicting the primary outcomes was mathematically assessed by the area under the curve (AUC)

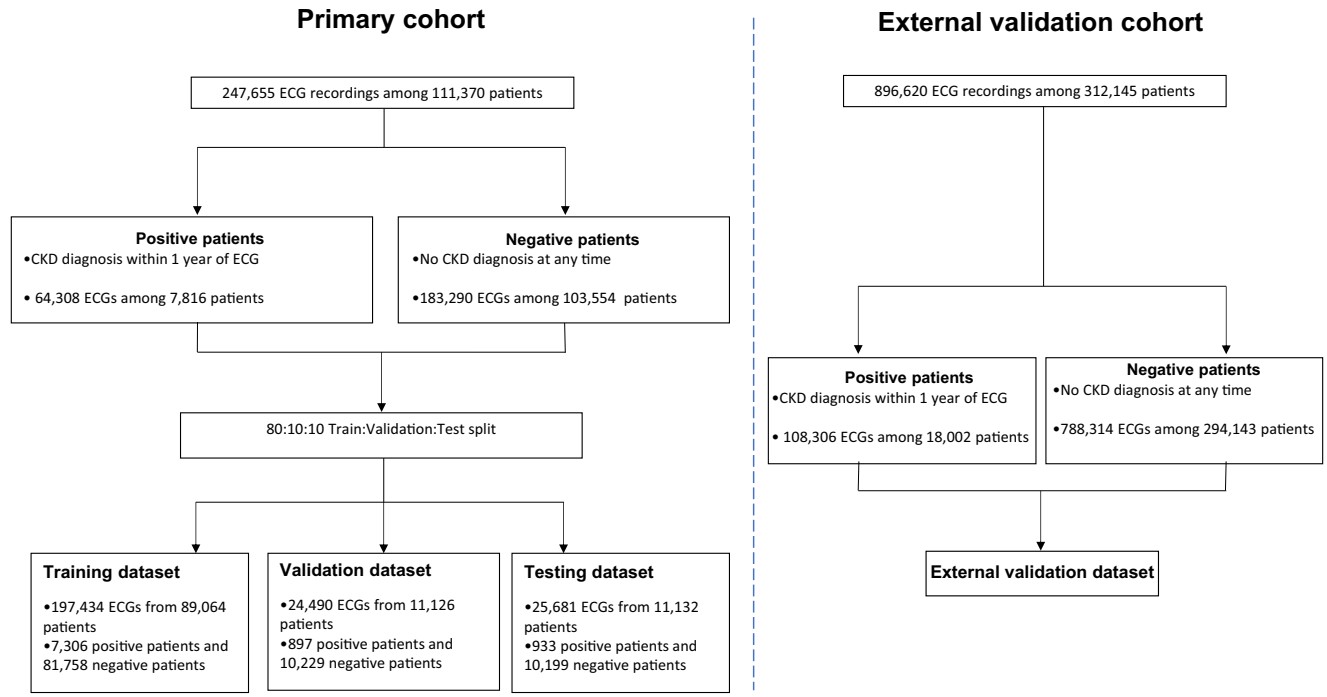

**Fig. 1 Study subject selection.** Our primary cohort consists of 111,370 patients and 247,655 ECGs between 2005 and 2019 from Cedars-Sinai Medical Center. The primary cohort was randomly split 8:1:1 into training, validation, and test cohorts. We also used 896,620 ECGs among 312,145 patients at Stanford Healthcare from 8/2005 to 6/2018 as external validation cohort. CKD Chronic kidney disease.

of the receiver operating characteristic (ROC) curve. After model derivation and training, primary and secondary analyses were performed on trained models using the held-out test cohort. Secondary sensitivity analyses were limited to procedures performed in patients with diabetes, hypertension, male, and age greater or lower than 60 years old. We computed two-sided 95% confidence intervals using 1000 bootstrapped samples for each calculation. Statistical analysis was performed in R and Python.

**Reporting summary**. Further information on research design is available in the Nature Portfolio Reporting Summary linked to this article.

## Results
**Primary cohort characteristics**. In total, we identified 17,860 patients with a CKD diagnosis at Cedars-Sinai Medical Center (7.8% of the total patient sample), among which 7816 had an ECG taken within a 1-year window of CKD diagnosis. Our primary cohort consisted of a total of 247,655 ECGs, of which 221,974 were randomized to the training set (for both training and validation) and 25,681 to the testing set. Of the primary cohort ECGs, 74.3% had no serum creatinine or eGFR estimation within 30 days and 50.7% of ECGs had no serum creatinine or eGFR estimation at any point in the EHR, however this does not capture outside hospital or paper clinic records of laboratory testing that might have been used in the diagnosis of CKD. The mean age of the primary cohort was $61.3 \pm 19.7$ years and 48% were female. Demographic and clinical characteristics are presented in Table 1. Demographics and clinical characteristics according to age group are presented in Supplementary Table 2.

**Model performance in the primary cohort**. Our 12-lead ECG-based model achieved discrimination of any stage CKD with an AUC of 0.767 (95% CI 0.76–0.773). The model performance was consistent across the range of CKD stage, with our model achieving an AUC of 0.753 (0.735–0.770) in discriminating mild CKD, AUC of 0.759 (0.750–0.767) in discriminating moderate-severe CKD, and AUC of 0.783 (0.773–0.793) in discriminating ESRD. In all cases, negative examples were defined as ECGs without CKD diagnoses.

Given the increased prevalence of wearable technologies, particularly devices that include single lead ECG information, we trained an additional deep learning model with information from only single lead ECG information to simulate the DLA's performance with single-lead wearable information. With 1-lead ECG waveform data, DLA achieved an AUC of 0.744 (0.737–0.751) in detecting any stage CKD, with sensitivity and specificity of 0.723 (0.723–0.723) and 0.643 (0.643–0.643), respectively.

Since early detection of CKD is crucial to prevent disease progression and complications in older age, we tested the performance of our model in younger patients (<60 years of age). 12-lead and 1-lead ECG-based DLAs were able to detect any stage CKD with AUCs of 0.843 (0.836–0.852) and 0.824 (0.815–0.832) among patients under 60 years of age, respectively.

We also tested the performance of our model separately among diabetic, hypertensive, older patients, who are generally considered as high-risk subgroups. 12-lead based model detected CKD with an AUC of 0.747 (0.707–0.783) among diabetic patients, an AUC of 0.711 (0.696–0.725) among patients with hypertension, and an AUC of 0.706 (0.697–0.716) among patients greater than 60 years old. When the model was trained with 12-lead ECG waveform, age, sex, diabetes, and hypertension, the model achieved similar discrimination of any stage CKD in the held-out test set with an AUC of 0.79 (0.781–0.798). Detailed results for 1-lead and 12-lead ECG-based DLA performance in the held-out test set are presented in Tables 2 and 3, while AUC curves are illustrated in Supplementary Fig. 1.

The model performed similarly in detecting CKD in subset populations of patients with albuminuria, patients with corresponding laboratory testing and documented eGFR, and in both ambulatory and in-hospital patients (Supplementary Table 3).

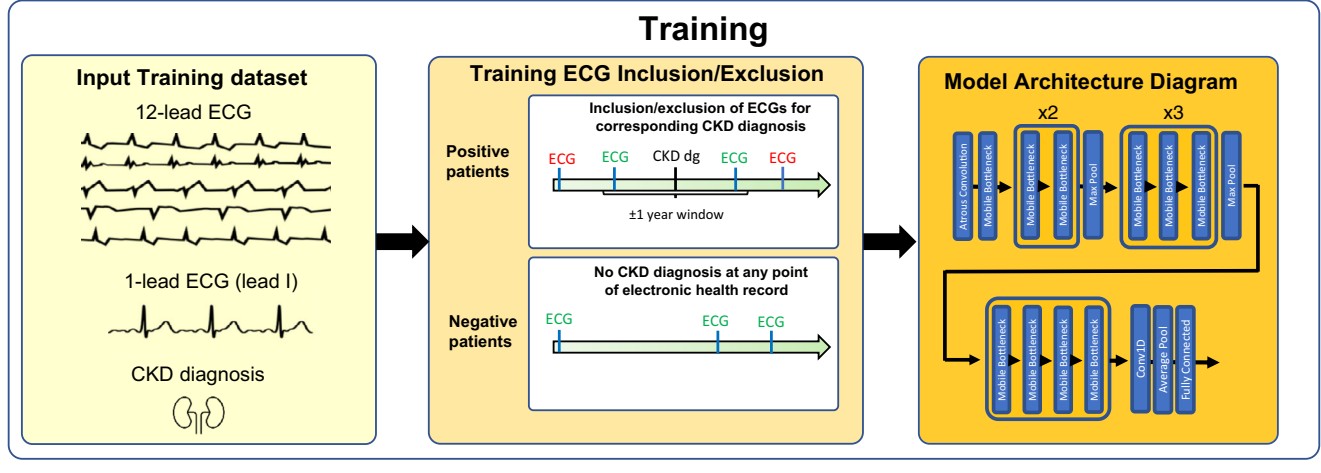

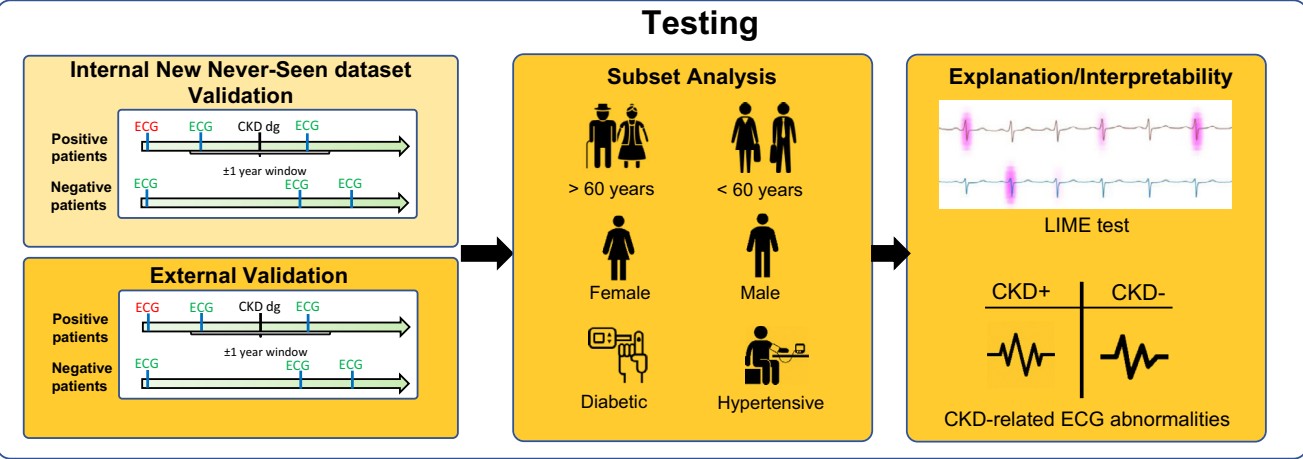

**Fig. 2 Schematic illustration of deep learning model training, testing, and validation.** We designed a convolutional neural network for ECG interpretation with potential for clinical data integration. The model was trained to predict CKD with the input of one 12-lead ECG within 1 year of CKD diagnosis. CKD Chronic kidney disease.

In patients with both a CKD diagnosis and eGFR estimated to be less than 60 mL/min, the AUC was 0.754 (0.737–0.771), and this performance was similar in patients with hyperkalemia with an AUC of 0.741 (0.698–0.787) and without hyperkalemia with an AUC of 0.758 (0.747–0.768). The model also performed well in patients with known albuminuria, with an AUC of 0.734 (0.723–0.745) and had similar performance regardless of the positive to negative ratio in the training set (Supplementary Table 4).

**Electrocardiographic features in CKD**. To understand the key features of relevance for our deep learning model to be able to detect CKD, we performed two sets of experiments to evaluate the ECG parameters that are important for identifying CKD. We found statistically significant differences in all available ECG variables (heart rate, PR interval, P wave duration, QRS duration, QTc interval, P-wave axis, R-wave axis, T-wave axis) between CKD stages (Supplementary Table 5).

Secondly, we used LIME to identify which ECG segments were particularly used in the identification of CKD. Supplementary Fig. 2 shows examples of LIME-highlighted ECG segments in 12-lead and 1-lead ECG waveforms taken from correctly recognized CKD and healthy control patients in the held-out test set. In both examples, the LIME-highlighted ECG features focused mostly on QRS complexes and PR intervals. In addition, QRS complexes and PR intervals in limb leads were most frequently highlighted, potentially denoting CKD-associated electrophysiologic alterations.

**External validation cohort characteristics**. The external validation cohort consisted of a total of 896,620 ECGs among 312,145 patients. The prevalence of mild CKD was 1.2% while 3.6% had moderate-severe CKD, and 0.9% had ESRD. The mean age of the external validation cohort was 56.7 ± 18.7 years and 50.4% were female. The proportion of Caucasians was 47.5%, while 3.6% were black, 12.3% were Asians, and 36.6% had other or unknown race. Demographic and clinical characteristics are presented in Table 1.

**Model performance in the external validation dataset**. In the external validation dataset, our 12-lead and 1-lead models' performances were comparable to the primary cohort. 12-lead ECG-based model achieved an AUC of 0.709 (0.708-0.710) in discriminating any stage CKD. 1-lead ECG-based model detected any stage CKD with an AUC of 0.701 (0.700–0.702).

Consistent with the primary cohort in which our model achieved higher CKD detection accuracy among younger patients, 12-lead and 1-lead ECG-based models achieved AUCs of 0.784 (0.782–0.786) and 0.777 (0.775–0.779) in detecting any stage CKD among subjects under 60 years of age, respectively. Detailed results for 1-lead and 12-lead ECG-based DLA performance in the external validation cohort are presented in Supplementary Tables 6 and 7.

**Discussion**

In the present study, we investigated the performance of a deep learning model to detect CKD using ECG waveforms.

**Table 1 Demographic and clinical characteristics in the internal and external dataset.**

| Characteristic | Internal training and validation datasets | | | External validation dataset |
|---|---|---|---|---|
| | Total | Training | Test | |
| Number of patients | 111,370 | 100,233 | 11,137 | 312,145 |
| Number of ECGs | 247,655 | 221,974 | 25,681 | 896,620 |
| Demographics | | | | |
| Age, years | 61.3 ± 19.7 | 61.3 ± 19.6 | 61.5 ± 19.9 | 56.7 ± 18.7 |
| Female, n (%) | 53,476 (48.0%) | 48,133 (48.0%) | 5,343 (48.0%) | 139,235 (50.4%) |
| BMI, kg/m$^2$ | 26.7 ± 16.6 | 26.7 ± 18.2 | 26.6 ± 7.2 | 27.6 ± 6.6 |
| Caucasian, n (%) | 67,253 (60.4%) | 60,550 (60.4%) | 6,703 (60.2%) | 148,367 (47.5%) |
| Black, n (%) | 15,323 (13.8%) | 13,751 (13.7%) | 1,572 (14.1%) | 11,191 (3.6%) |
| Asian, n (%) | 6,135 (5.5%) | 5,558 (5.5%) | 577 (5.2%) | 38,229 (12.3%) |
| Other/unknown race, n (%) | 22,659 (20.3%) | 20,374 (20.3%) | 2,285 (20.5%) | 114,358 (36.6%) |
| Clinical characteristics, n (%) | | | | |
| Hypertension | 25,446 (26.4%) | 22,623 (26.4%) | 2,823 (26.3%) | 110,311 (35.3%) |
| Diabetes Mellitus | 8,728 (7.8%) | 6,970 (7.8%) | 914 (8.2%) | 14,152 (4.5%) |
| Cardiovascular disease | 15,719 (14.1%) | 12,561 (14.1%) | 1,577 (14.2%) | 34,582 (11.1%) |
| Heart Failure | 9,604 (8.6%) | 7,634 (8.6%) | 995 (8.9%) | 20,167 (6.5%) |
| Proteinuria | 1,007 (0.9%) | 795 (0.9%) | 110 (1.0%) | 2,086 (0.7%) |
| Anemia | 11,933 (10.7%) | 9,537 (10.7%) | 1,194 (10.7%) | 18,653 (6.0%) |
| Chronic kidney disease, n (%) | | | | |
| Mild (Stage 1-2) | 864 (0.8%) | 782 (0.8%) | 82 (0.7%) | 3,727 (1.2%) |
| Moderate-severe (Stage 3-5) | 3918 (3.5%) | 3515 (3.5%) | 403 (3.6%) | 11,335 (3.6%) |
| ESRD | 3034 (2.7%) | 2722 (2.7%) | 312 (2.8%) | 2,940 (0.9%) |
| eGFR*, ml/min/1.73 m$^2$ | | | | |
| <15 | 1,744 (7.0%) | 1,531 (7.0%) | 213 (7.7%) | |
| 15-29 | 1,789 (7.2%) | 1,563 (7.1%) | 226 (2.0%) | |
| 30-60 | 5,477 (22.1%) | 4,831 (22.0%) | 646 (23.3%) | |
| >60 | 15,575 (62.9%) | 13,904 (63.3%) | 1,671 (60.3%) | |
| Potassium (mmol/L) | | | | |
| >5.5 | 2,356 (2.7%) | 2,075 (2.7%) | 281 (2.9%) | |
| <=5.5 | 85,573 (97.3%) | 76,106 (97.3%) | 9,467 (97.1%) | |

Continuous variables are presented as mean±standard deviation. *within a month of ECG (data available from 24,585 patients). *BMI* Body mass index, *eGFR* estimated glomerular filtration rate, *ESRD* End-stage renal disease.

**Table 2 Performance of the 12-lead ECG-based deep learning algorithm in the internal dataset.**

| Model | AUC (95% CI) | Sensitivity (95% CI) | Specificity (95% CI) | PPV (95% CI) | NPV (95% CI) |
|---|---|---|---|---|---|
| 12-lead ECG models | | | | | |
| Any stage CKD | 0.767 (0.76-0.773) | 0.699 (0.699-0.699) | 0.698 (0.698-0.698) | 0.443 (0.443-0.443) | 0.871 (0.871-0.871) |
| Mild CKD | 0.753 (0.735-0.77) | 0.75 (0.75-0.75) | 0.644 (0.644-0.644) | 0.064 (0.064-0.064) | 0.987 (0.987-0.987) |
| Moderate-severe CKD | 0.759 (0.75-0.767) | 0.785 (0.785-0.785) | 0.598 (0.598-0.598) | 0.271 (0.271-0.271) | 0.936 (0.936-0.936) |
| ESRD | 0.783 (0.773-0.793) | 0.704 (0.704-0.704) | 0.726 (0.726-0.726) | 0.237 (0.237-0.237) | 0.953 (0.953-0.953) |
| High Risk Subgroup analyses for any stage CKD | | | | | |
| Diabetic patients | 0.747 (0.707-0.783) | 0.699 (0.699-0.699) | 0.682 (0.682-0.682) | 0.906 (0.906-0.906) | 0.342 (0.342-0.342) |
| Hypertensive patients | 0.711 (0.696-0.725) | 0.648 (0.629-0.668) | 0.662 (0.644-0.68) | 0.638 (0.618-0.658) | 0.672 (0.653-0.691) |
| Age > 60 years | 0.706 (0.697-0.716) | 0.604 (0.604-0.604) | 0.701 (0.701-0.701) | 0.397 (0.397-0.397) | 0.844 (0.844-0.844) |
| Male | 0.764 (0.755-0.772) | 0.727 (0.727-0.727) | 0.666 (0.666-0.666) | 0.485 (0.485-0.485) | 0.849 (0.849-0.849) |
| Female | 0.756 (0.745-0.768) | 0.699 (0.699-0.699) | 0.680 (0.680-0.680) | 0.351 (0.351-0.351) | 0.902 (0.902-0.902) |
| Screening cohort (age <60) | | | | | |
| Any stage CKD | 0.843 (0.836-0.852) | 0.761 (0.761-0.761) | 0.787 (0.787-0.787) | 0.57 (0.57-0.57) | 0.899 (0.899-0.899) |
| Mild CKD | 0.795 (0.766-0.823) | 0.702 (0.702-0.702) | 0.748 (0.748-0.748) | 0.075 (0.075-0.075) | 0.989 (0.989-0.989) |
| Moderate-severe CKD | 0.854 (0.842-0.865) | 0.792 (0.792-0.792) | 0.787 (0.787-0.787) | 0.373 (0.373-0.373) | 0.959 (0.959-0.959) |
| ESRD | 0.842 (0.831-0.853) | 0.746 (0.746-0.746) | 0.791 (0.791-0.791) | 0.394 (0.394-0.394) | 0.945 (0.945-0.945) |

*AUC* area under the receiver operating characteristics curve, *CI* Confidence interval, *CKD* Chronic kidney disease, *ESRD* End-stage renal disease, *PPV* positive predictive value, *NPV* negative predictive value.

Our 12-lead ECG-based model had good accuracy in identifying any stage CKD and higher accuracy in detecting CKD in patients under 60 years of age. Accuracy also improved along with the worsening CKD stage. These results were validated in a separate health care system, that also showed good discrimination accuracy for the presence of any stage CKD in the whole study population and higher discrimination accuracy among patients under 60 years of age. While 12-lead ECGs are widely available in the healthcare unit settings, rapid adoption of wearable technology has also introduced opportunities for large-scale data collection outside of formal healthcare settings. Our 1-lead ECG-based DLA showed good discrimination accuracy for CKD in young patients, suggesting artificial intelligence may possess significant potential in widescale screening in this patient

**Table 3 Performance of the 1-lead ECG-based deep learning algorithm in the internal dataset.**

| Model | AUC (95% CI) | Sensitivity (95% CI) | Specificity (95% CI) | PPV (95% CI) | NPV (95% CI) |
|---|---|---|---|---|---|
| 1-lead ECG models | | | | | |
| Any stage CKD | 0.744 (0.737–0.751) | 0.723 (0.723–0.723) | 0.643 (0.643–0.643) | 0.41 (0.41–0.41) | 0.871 (0.871–0.871) |
| Mild CKD | 0.746 (0.728–0.764) | 0.735 (0.735–0.735) | 0.66 (0.66–0.66) | 0.066 (0.066–0.066) | 0.987 (0.987–0.987) |
| Moderate-severe CKD | 0.735 (0.726–0.744) | 0.732 (0.732–0.732) | 0.618 (0.618–0.618) | 0.267 (0.267–0.267) | 0.924 (0.924–0.924) |
| ESRD | 0.757 (0.748–0.767) | 0.738 (0.738–0.738) | 0.647 (0.647–0.647) | 0.202 (0.202–0.202) | 0.953 (0.953–0.953) |
| High Risk Subgroup analyses for any stage CKD | | | | | |
| Diabetic patients | 0.663 (0.625–0.707) | 0.819 (0.819–0.819) | 0.457 (0.457–0.457) | 0.868 (0.868–0.868) | 0.367 (0.367–0.367) |
| Hypertensive patients | 0.684 (0.668–0.699) | 0.658 (0.639–0.677) | 0.614 (0.594–0.633) | 0.61 (0.589–0.629) | 0.662 (0.641–0.68) |
| Age > 60 years | 0.681 (0.671–0.691) | 0.679 (0.679–0.679) | 0.588 (0.588–0.588) | 0.35 (0.35–0.35) | 0.849 (0.849–0.849) |
| Male | 0.742 (0.733–0.75) | 0.747 (0.747–0.747) | 0.612 (0.612–0.612) | 0.455 (0.455–0.455) | 0.848 (0.848–0.848) |
| Female | 0.735 (0.723–0.747) | 0.672 (0.672–0.672) | 0.681 (0.681–0.681) | 0.342 (0.342–0.342) | 0.894 (0.894–0.894) |
| Screening cohort (age <60) | | | | | |
| Any stage CKD | 0.824 (0.815–0.832) | 0.812 (0.812–0.812) | 0.705 (0.705–0.705) | 0.506 (0.506–0.506) | 0.91 (0.91–0.91) |
| Mild CKD | 0.819 (0.791–0.844) | 0.771 (0.771–0.771) | 0.758 (0.758–0.758) | 0.085 (0.085–0.085) | 0.991 (0.991–0.991) |
| Moderate-severe CKD | 0.828 (0.816–0.84) | 0.814 (0.814–0.814) | 0.713 (0.713–0.713) | 0.313 (0.313–0.313) | 0.96 (0.96–0.96) |
| ESRD | 0.82 (0.808–0.831) | 0.82 (0.82–0.82) | 0.693 (0.693–0.693) | 0.328 (0.328–0.328) | 0.955 (0.955–0.955) |

*AUC* area under the receiver operating characteristics curve, *CI* Confidence interval, *CKD* Chronic kidney disease, *ESRD* End-stage renal disease. *PPV* positive predictive value. *NPV* negative predictive value.

population. One-lead ECGs could also increase screening rates in high-risk patients (Supplementary Figs. 3 and 4). However, the integration of artificial intelligence in electronic devices requires a more detailed evaluation of accuracy in a real-life setting.

Low awareness of CKD and limitations in current screening measures highlight the urgency of novel screening strategies to increase detection rates of early-stage CKD. Being non-invasive and often obtained in the clinic, ECGs are often the first line of clinical evaluation. In our healthcare system, 74% of ECGs obtained did not have laboratory testing of kidney function within 30 days. Previous studies have demonstrated that the cost-effectiveness of CKD screening is highly dependent on patient risk factor profile and CKD probability, and there has been debate on whether CKD screening should be targeted only to high-risk patients, or also extend to patients without risk factors for CKD[22–25]. Although screening high-risk patients is guideline-recommended, testing rates remain low as only about 20% of high-risk patients receive guideline-recommended assessment in the U.S.[26]. Consequently, most of the high-risk patients are likely to be unaware of underlying CKD[2,3]. Moreover, a substantial proportion of all CKD patients are not high-risk patients and hence not recommended to be screened regularly, which further highlights the need for novel screening methods.

Our model performed better at detecting CKD in younger patients, whereas detection accuracy was lower in older and high-risk patients. Reasons for this observation are not fully clear but may be due to the fact that younger patients in general have fewer comorbidities, meaning that any detected ECG abnormalities may be especially meaningful and specific. Although older age is a well-known risk marker for CKD, the prevalence of CKD in younger patients is also notably high in the U.S. (8–10% in <65 years)[3]. Remarkably, however, awareness of underlying CKD is also very low in younger patients, as only about 8% are aware of the disease[3]. Given the availability of effective low-risk CKD treatments and the reversibility of CKD, there are substantial potential benefits for detecting and treating CKD, especially in the young. A recent paper by Kwon et al.[27] also used data from ECG waveforms in addition to age and sex to develop a DLA to detect changes in eGFR, which can include both patients with acute kidney injury (e.g., dehydration, pharmacotherapy, urinary tract obstruction) as well as chronic kidney disease. Their model achieved a slightly higher performance with an AUC of 0.86–0.91, however reaffirms the overall conclusion that renal abnormalities

can be detected by CKD within large cohorts across multiple international sites.

The strengths of our study include the large cohort of patients undergoing ECG recording across a decade and the use of state-of-the-art deep learning architectures. We also used two separate approaches to understand the key features of relevance for our deep learning model. While previous studies have reported that patients with CKD have high rates of P wave abnormalities, prolonged PR interval, QTc prolongation, QT dispersion, and left ventricle hypertrophy[28–31], in the present study CKD was associated with skewed P-, R-, and T-wave axes in addition to prolonged QRS, PR, and QTc intervals. However, a few limitations warrant consideration. Our study is retrospective, and study populations are derived from two large academic medical centers situated in dense urban metropolitan areas using ICD-9 codes. By prioritizing priority codes, we sought to avoid incidences of acute rather than chronic kidney injury, however we cannot exclude the possibility that some of the study subjects without CKD diagnosis in electronic health records have an undiagnosed disease, as especially mild-stage CKD can often be undiagnosed, particularly using an ICD9 code-based adjudication. In the subset with both ICD-9 code adjudication of CKD as well as laboratory testing, the ICD-9 codes were consistent with the calculated eGFR, however only a minority of patients were able to be linked to data regarding microalbuminuria.

Validation in prospective general population cohorts in outpatient settings is required to confirm an ECG-based DLA's ability to recognize patients with CKD. Although the prevalence of CKD was low in our training cohort with ECGs, and this prevalence not directly comparable to epidemiological cohorts with CKD (as ECGs are more commonly obtained in patients without CKD), we show that our disease definitions is consistent with laboratory testing and documented eGFR (Supplementary Table 8) and that our deep learning approach is relatively insensitive in model accuracy to disease prevalence in the training set (Supplementary Table 4). The prevalence of hypertension diagnosis may be underestimated in the internal cohort, however our model performed similarly well in internal and external test cohorts with different prevalences of hypertension.

By 2030, the UN's Sustainable Development Goals are to reduce premature mortality related to non-communicable diseases by a third. Given the high prevalence of asymptomatic CKD, serious consequences of untreated disease, presence of

effective low-risk treatment, and detectable preclinical state with inexpensive and simple diagnostic tests, CKD represents a good target for large-scale population screening and harbors the potential for reducing premature mortality related to non-communicable diseases. In addition to the high mortality and morbidity due to CKD, treatment costs for CKD are also high and have increased during the last decades[32]. Especially, the increasing number of patients requiring renal replacement creates challenges for health care systems worldwide, and the shortage of sufficient replacement services may cause at least 2 million premature deaths annually[33]. Therefore, widely available, inexpensive, and effective CKD prevention and management strategies are warranted to enable equal opportunities in reducing CKD-related disability-adjusted life years.

## Conclusions

Our ECG-based deep learning model was able to detect CKD with good discrimination accuracy in multiple study populations and with particularly high accuracy in patients under 60 years of age. These results suggest that deep learning-based ECG analysis may provide additional value in detecting various CKD stages, especially in younger patients. The clinical significance of this study lies in the potential enhancement of screening methods for the early detection of CKD, which is crucial to enable early treatment and prevent disease progression.

## Data availability

All analytical methods applied for the deep learning algorithm are included in this published article, supplementary files. The patient data is not publicly available due to potentially identifiable nature of the associated data. De-identified data is available from the corresponding author on reasonable request.

## Code availability

Code is available at https://github.com/ecg-net/CKDscreening[34].

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

## Acknowledgements

This work was supported in part by the National Institutes of Health NIH K99 HL157421-01. L.H. was supported by Sigrid Juselius Foundation, The Finnish Cultural Foundation, Instrumentarium Science Foundation, Orion Research Foundation, and Paavo Nurmi Foundation. The funding sources had no involvement in the preparation of this work or the decision to submit for publication.

## Author contributions

Study design and conception: L.H., M.C., S.S.C., and D.O. Acquisition, analysis or interpretation of data: L.H., M.C., N.Y., J.W.H., J.T., M.J., P.F., A.K., R.K.S., J.E., S.C., J.Z., S.S.C., and D.O. Drafting of the manuscript: L.H., M.C., and D.O. Statistical analysis: M.C., J.W.H., and D.O. Critical revision of the manuscript for important intellectual content: N.Y., J.W.H., J.T., M.J., P.F., A.K., R.K.S., J.E., S.C., J.Z., and S.S.C. Administrative and material support: S.S.C. and D.O. Obtained funding: D.O. Supervision: S.S.C. and D.O. Full access to the data: M.C. and D.O. All authors reviewed and approved the final version of the manuscript.

## Competing interests

The authors declare no competing interests.
