## [Peer Review File · Communications Medicine]

Reviewers' comments:

Reviewer #1 (Remarks to the Author):

The authors proposed a deep learning model for the automatic screening of chronic kidney disease based on electrocardiographic recordings. The manuscript is well designed and written. Experimental data, figures, and results tables are clear to easily understand. However, I have some recommendations below.

- There are some similar studies (doi:10.1001/jamacardio.2019.0640, doi:10.2196/34724) published, and the authors should be shown the differences, and outstanding, advantages of the proposed DL model.
- another thing is that if possible, the authors should be given more information about the hardware and software spec for the DL model training and testing for the followers.

Reviewer #2 (Remarks to the Author):

In their study, Holmstrom et al seek to answer an important medical question – can an easily accessible screen for chronic kidney disease be identified. As the authors accurately note, such a test would be clinically important in enabling early therapy to prevent significant morbidity and mortality. In the present work the investigators apply a convolutional neural network to a standard 12 lead ECG to determine the absence or presence of renal disease in a retrospective cohort, and find the overall test AUC is 0.767, with greater performance in younger cohorts.

The application of convolution neural networks to the 12 lead ECG for several years has been a known powerful screen in retrospective studies for cardiac and non-cardiac disease, and has been demonstrated to apply across diverse, global populations.¹⁻¹⁴ In more recent and novel work, the approach has been shown to identify occult disease in prospective studies, and to correlated to therapeutic drug effects.^{15,16}

The specific question addressed by Holmstrom et al is important and has been recently address by Kwon et al, using similar techniques (retrospective analysis using convolutional neural networks).¹⁷ Interestingly, Kwon et al found an AUC 0.9, substantially higher than the findings of Holmstrom et al. On reason for this might be the much shorter time interval between the ECG and renal assessment (30 minutes for Kwon et al, vs. up to a year in the report by Holmstrom). Holmstrom et al could further clarify their work by addressing the following:

1. How does their work compare to that of Kwon et al, and is the test performance so different?
2. Were outpatient and inpatient ECGs used? Inpatient ECGs may have greater dynamic changes due to conditions beyond those being studied, and may be more susceptible to the long interval between ECG and renal assessment. Please comment
3. Might other variables in addition to the ECG strengthen model performance?
4. Were there differences in performance based on electrolyte abnormalities or mechanism

of renal injury? Of race, sex or ethnicity?

5. Modeling the use of the test – such as PPV, NPV etc would be useful, as the low PPV (particularly in young cohorts) may limit real-world utilization. I did not see this information in the report

In summary, Halmstrom et al present important confirmatory work of previously published research using well established techniques. As such the work would be meritorious for publication in a nephrology or electrocardiography journal, but lacks the innovation or novelty in analytic techniques or study design, or prospective clinical application to warrant publication in a high impact journal such as Nature.

References:

- 1 Ahn, J. C. et al. Development of the AI-Cirrhosis-ECG Score: An Electrocardiogram-Based Deep Learning Model in Cirrhosis. *Am J Gastroenterol* 117, 424-432 (2022).
<https://doi.org/10.14309/ajg.0000000000001617>
- 2 Attia, I. Z. et al. External validation of a deep learning electrocardiogram algorithm to detect ventricular dysfunction. *Int J Cardiol* 329, 130-135 (2021).
<https://doi.org/10.1016/j.ijcard.2020.12.065>
- 3 Attia, Z. I. et al. Age and Sex Estimation Using Artificial Intelligence From Standard 12-Lead ECGs. *Circ Arrhythm Electrophysiol* 12, e007284 (2019).
<https://doi.org/10.1161/CIRCEP.119.007284>
- 4 Attia, Z. I. et al. Screening for cardiac contractile dysfunction using an artificial intelligence-enabled electrocardiogram. *Nat Med* 25, 70-74 (2019). <https://doi.org/10.1038/s41591-018-0240-2>
- 5 Attia, Z. I. et al. An artificial intelligence-enabled ECG algorithm for the identification of patients with atrial fibrillation during sinus rhythm: a retrospective analysis of outcome prediction. *The Lancet* 394, 861-867 (2019). [https://doi.org/10.1016/s0140-6736\(19\)31721-0](https://doi.org/10.1016/s0140-6736(19)31721-0)
- 6 Galloway, C. D. et al. Development and Validation of a Deep-Learning Model to Screen for Hyperkalemia From the Electrocardiogram. *JAMA Cardiol* 4, 428-436 (2019).
<https://doi.org/10.1001/jamacardio.2019.0640>
- 7 Cohen-Shelly, M. et al. Electrocardiogram screening for aortic valve stenosis using artificial intelligence. *Eur Heart J* 42, 2885-2896 (2021). <https://doi.org/10.1093/eurheartj/ehab153>
- 8 Attia, Z. I., Harmon, D. M., Behr, E. R. & Friedman, P. A. Application of artificial intelligence to the electrocardiogram. *Eur Heart J* 42, 4717-4730 (2021).
<https://doi.org/10.1093/eurheartj/ehab649>
- 9 Adedinsowo, D. et al. Artificial Intelligence-Enabled ECG Algorithm to Identify Patients With Left Ventricular Systolic Dysfunction Presenting to the Emergency Department With Dyspnea. *Circ Arrhythm Electrophysiol* 13, e008437 (2020).
<https://doi.org/10.1161/CIRCEP.120.008437>
- 10 Adedinsowo, D. A. et al. Detecting cardiomyopathies in pregnancy and the postpartum period with an electrocardiogram-based deep learning model. *Eur Heart J Digit Health* 2, 586-596 (2021). <https://doi.org/10.1093/ehjdh/ztab078>
- 11 Harmon, D. M. et al. Real-world performance, long-term efficacy, and absence of bias in

the artificial intelligence enhanced electrocardiogram to detect left ventricular systolic dysfunction. *European Heart Journal - Digital Health* 3, 238-244 (2022).

<https://doi.org:10.1093/ehjdh/ztac028>

12 Ribeiro, A. H. et al. Automatic diagnosis of the 12-lead ECG using a deep neural network. *Nat Commun* 11, 1760 (2020). <https://doi.org:10.1038/s41467-020-15432-4>

13 Vaid, A. et al. Using Deep-Learning Algorithms to Simultaneously Identify Right and Left Ventricular Dysfunction From the Electrocardiogram. *JACC: Cardiovascular Imaging* 15, 395-410 (2022). <https://doi.org:https://doi.org/10.1016/j.jcmg.2021.08.004>

14 Zhu, H. et al. Automatic multilabel electrocardiogram diagnosis of heart rhythm or conduction abnormalities with deep learning: a cohort study. *Lancet Digit Health* 2, e348-e357 (2020). [https://doi.org:10.1016/S2589-7500\(20\)30107-2](https://doi.org:10.1016/S2589-7500(20)30107-2)

15 Yao, X. et al. Artificial intelligence-enabled electrocardiograms for identification of patients with low ejection fraction: a pragmatic, randomized clinical trial. *Nat Med* 27, 815-819 (2021). <https://doi.org:10.1038/s41591-021-01335-4>

16 Tison Geoffrey, H. et al. Assessment of Disease Status and Treatment Response With Artificial Intelligence–Enhanced Electrocardiography in Obstructive Hypertrophic Cardiomyopathy. *Journal of the American College of Cardiology* 79, 1032-1034 (2022). <https://doi.org:10.1016/j.jacc.2022.01.005>

17 Kwon, J. M. et al. Artificial intelligence assessment for early detection and prediction of renal impairment using electrocardiography. *Int Urol Nephrol* 54, 2733-2744 (2022). <https://doi.org:10.1007/s11255-022-03165-w>

Reviewer #3 (Remarks to the Author):

In this paper, the authors assessed the predictive value of machine learning algorithms based on ECG changes to detect chronic kidney disease (CKD). Model performance was reasonable, especially in patients below 60 years of age. Especially as a proof of concept study, the data are interesting

1. Whereas it is quite relevant that ECG changes are associated with even mild CKD, this has been shown in previous studies (e.g. Kestenbaum CJASN 2007) the practical consequence of the study remains somewhat elusive in the paper. This because CKD can be detected in a relatively straightforward and non-expensive manner (blood and urine tests) which is already part of guidelines for high risk groups.

Most likely, costs for ECG will be higher, not counting potential additional costs of using a deep learning algorithm, whereas PPV is relatively low . It would be interesting to address how the use of ECG can overcome current limitations in screening for occult CKD , other than the fact that it may be advisable to measure kidney function in patients with abnormal results on a ECG that was performed for another reason? Do the authors imply to use ECG as a primary screening tool or as an add-on , e.g. in patients who are not under medical care but do have a Smartwatch with lead I ECG available?

2. In the primary cohort characteristics (p6), 17860 patients were diagnosed with CKD, but 51% of (patients with?) ECGs did not have any measurement of renal function in the system. How was the diagnosis of CKD confirmed in these patients. Were those patients (diagnosis

CKD, but no data on renal function) included in the analysis?

3. In the same paragraph, it is mentioned that 50.7% of ECGs had no serum creatinine or GFR estimations, which should be rephrased. Does this mean "50.7 of patients with ECGs" or is the patient number lower given the fact that patients might have undergone multiple ECG measurements?

4. Were patients who did not have any information on eGFR but underwent ECG included in the analysis? If so, would it not be better to include only patients in whom renal function measurements (either indicative of CKD or not) in the analysis? This because one cannot be sure that the patients without CKD diagnosis or ECG measurements indeed have a normal renal function, given the high prevalence of CKD in the general population?

5. In table 1, the classification of CKD (moderate/end stage) i.e. 6952 patients does not seem to match with eGFR criteria where 9010 patients had a eGFR <60 ml/min/1.73m². The prevalence of ESRD appears to be very high (2.7% in a population study!) as compared to the other stages. In the table, stage 5 CKD is included in the "moderate" category, which is incorrect. The fact that a different type of percentages are mentioned for the eGFR (adding up to 99,2%) as compared to the stages of CKD (where they indicate percentages of the population) is somewhat confusing

Reviewers' comments:

Reviewer #1 (Remarks to the Author):

The authors proposed a deep learning model for the automatic screening of chronic kidney disease based on electrocardiographic recordings. The manuscript is well designed and written. Experimental data, figures, and results tables are clear to easily understand. However, I have some recommendations below.

- There are some similar studies (doi:10.1001/jamacardio.2019.0640, doi:10.2196/34724) published, and the authors should be shown the differences, and outstanding, advantages of the proposed DL model.

Reply: Thank you for these thoughtful comments and we agree. While hyperkalemia can occur in the presence of chronic kidney disease, it is not common or prevalent in all patients with CKD, and hence unlikely to be the primary signal detected by our ML model. In the JAMA Cardiology paper by Galloway et al, in their cohort of patients with CKD, the prevalence of hyperkalemia was between 2.6% and 4.8%. In new analyses in this revision, for additional support of our conclusions, we add additional information on the mean K level at the time of each ECG, showing that the majority of patients had normal K values at time of ECG (97.3%) (Table 1), and in additional new experiments, we show that model performance did not improve in the subset with hyperkalemia, rather decreased (AUC of 0.741) compared to an AUC of 0.758 when normal potassium, suggesting the model is picking up additional information distinct and unique from hyperkalemia. In this revision, we also add additional discussion on the signal of hyperkalemia adding to CKD detection.

Page 4 line 94 (Introduction): *In addition to myocardial remodeling, CKD associates with a variety of electrolyte abnormalities that also cause widespread ECG abnormalities (e.g., decreased T-wave amplitudes in hypokalemia, large-amplitude T-waves, and prolonged QRS duration in hyperkalemia, and QTc prolongation in hypocalcemia)(18). Prior work has shown such patterns are detectable on ECG waveforms, contributing to the AI-ECG detection of hyperkalemia, which might augment a model's ability to detect CKD (19,20). However, given the relative infrequency of overt abnormalities, likely not the primary feature analyzed in detecting CKD. Given such observations, it may be possible that asymptomatic CKD presents with subtle ECG alterations that are not visible to the human eye.*

Characteristic	Internal training and validation datasets			External validation dataset
	Total	Training	Test	
Number of patients	111,370	100,233	11,137	312,145
Number of ECGs	247,655	221,974	25,681	896,620
Demographics				
Age, years	61.3 ±19.7	61.3 ±19.6	61.5 ±19.9	56.7±18.7
Female, n (%)	53,476 (48.0%)	48,133 (48.0%)	5,343 (48.0%)	139,235 (50.4%)
BMI, kg/m ²	26.7±16.6	26.7 ±18.2	26.6 ±7.2	27.6±6.6
Caucasian, n (%)	67,253 (60.4%)	60,550 (60.4%)	6,703 (60.2%)	148,367 (47.5%)
Black, n (%)	15,323 (13.8%)	13,751 (13.7%)	1,572 (14.1%)	11,191 (3.6%)
Asian, n (%)	6,135 (5.5%)	5,558 (5.5%)	577 (5.2%)	38,229 (12.3%)
Other/unknown race, n (%)	22,659 (20.3%)	20,374 (20.3%)	2,285 (20.5%)	114,358 (36.6%)
Clinical characteristics, n (%)				
Hypertension	25,446 (26.4%)	22,623 (26.4%)	2,823 (26.3%)	110,311 (35.3%)
Diabetes Mellitus	8,728 (7.8%)	6,970 (7.8%)	914 (8.2%)	14,152 (4.5%)
Cardiovascular disease	15,719 (14.1%)	12,561 (14.1%)	1,577 (14.2%)	34,582 (11.1%)
Heart Failure	9,004 (8.0%)	7,034 (8.0%)	995 (8.9%)	20,107 (6.5%)
Proteinuria	1,007 (0.9%)	795 (0.9%)	110 (1.0%)	2,086 (0.7%)
Anemia	11,933 (10.7%)	9,537 (10.7%)	1,194 (10.7%)	18,653 (6.0%)
Chronic kidney disease, n (%)				
Mild (Stage 1-2)	864 (0.8%)	782 (0.8%)	82 (0.7%)	3,727 (1.2%)
Moderate-severe (Stage 3-5)	3918 (3.5%)	3515 (3.5%)	403 (3.6%)	11,335 (3.6%)
ESRD	3034 (2.7%)	2722 (2.7%)	312 (2.8%)	2,940 (0.9%)
eGFR*, ml/min/1.73 m²				
<15	1,744 (7.0%)	1,531 (7.0%)	213 (7.7%)	
15-29	1,789 (7.2%)	1,563 (7.1%)	226 (2.0%)	
30-60	5,477 (22.1%)	4,831 (22.0%)	646 (23.3%)	
>60	15,575 (62.9%)	13,904 (63.3%)	1,671 (60.3%)	
Potassium (mmol/L)				
>5.5	2,356 (2.7%)	2,075 (2.7%)	281 (2.9%)	
<=5.5	85,573 (97.3%)	76,106 (97.3%)	9,467 (97.1%)	

Table 1. Demographic and clinical characteristics in the internal and external dataset. Continuous variables are presented as mean±standard deviation. *within a month of ECG (data available from 24,585 patients). BMI=Body mass index, eGFR=estimated glomerular filtration rate, ESRD=End-stage renal disease.

Test task	AUC (95% CI)
eGFR<60 ml/min/1.73 m ² *	0.754 (0.737-0.771)
Albuminuria**	0.734 (0.723 - 0.745)
CKD among ambulatory patients	0.763 (0.74-785)
CKD among in-hospital patients	0.762 (0.752-0.772)
CKD among patients with K >5.5mmol/L***	0.741 (0.698 - 0.787)
CKD among patients with K <=5.5mmol/L***	0.758 (0.747 - 0.768)

White patients	0.764 (0.752 - 0.777)
Black patients	0.76 (0.738 - 0.779)
Asian patients	0.749 (0.715 - 0.782)
Unknown or unspecified race patients	0.769 (0.74 - 0.802)

Supplemental Table 3. 12-lead ECG based deep learning model's performance in additional tests in the internal cohort. *within a month of ECG (data available from 23,799 patients), **albumin-to-creatinine ratio >30mg/g (data available from 7,218 patients), ***within a month of ECG. AUC=area under the receiver operating characteristics curve, CI=Confidence interval, CKD=chronic kidney disease, eGFR=estimated glomerular filtration rate.

- another thing is that if possible, the authors should be given more information about the hardware and software spec for the DL model training and testing for the followers.

Reply: Good point and we agree. To clarify the model training, we have now provided more information about the DL model training and testing *in the Supplemental material*.

Page 5 line 138 (Methods): We designed a novel convolutional neural network, for ECG interpretation with potential for clinical data integration to predict the primary outcomes of chronic kidney disease and end-stage renal disease (Figure 2). The model was trained to predict outcomes with the input of one 12-lead ECG obtained within 1 year of diagnosis. Please see *Supplementary Materials* for additional details on model training.

Supplemental Methods: Deep Learning Algorithm Training

Based on prior literature regarding lightweight deep learning model architecture design and neural architecture search^{1,2}, our deep learning model was designed to analyze 12-lead ECG waveform data starting with atrous convolutions followed by subsequent multi-channel 1D convolutions. The number of layers paralleled the design of EfficientNet³, and to optimize model runtime and minimize model complexity, the number of layers were limited to less than 1/10th the size of previously described architectures.^{3,5} After initial atrous layers, the model incorporated convolutional layers with an inverted residual structure where the input and output are bottleneck 9 layers with an intermediate expansion layer.¹ In each set of expansion layers with bottleneck layers preceding and succeeding, the number of input channels gradually increased to allow for integration of information across ECG leads. This model was previously used in prior work on ECG deep learning predicting other tasks⁶ with code available online.

Model training was performed on a Linux (Ubuntu) computer with an 24 core AMD Threadripper 2960X and two RTX 2090s with a total of 48GB VRAM. The model was

initialized with random weights and trained with a loss function of binary cross entropy for 100 epochs using an ADAM optimizer with an initial learning rate between 5e-3 and 1e-4. Early stopping was performed based on validation dataset's area under the receiver operating curve. The atrous convolution's dilation and step size was grid-searched by hyperparameter tuning for optimal AUC with all other hyperparameters held constant (Supplemental Figure 1). Local Interpretable Model-agnostic Explanations³⁵ was used to identify and visualize relevant features in the ECG used for model decision making.

1. Sandler M, Howard A, Zhu M, Zhmoginov A and Chen L-C. MobileNetV2: Inverted Residuals and Linear Bottlenecks. *arXiv [csCV]*. 2018.
2. Tan M and Le QV. EfficientNet: Rethinking Model Scaling for Convolutional Neural Networks. *arXiv [csLG]*. 2019.
3. Davis C, Tait G, Carroll J, Wijeyesundera DN and Beattie WS. The Revised Cardiac Risk Index in the new millennium: a single-centre prospective cohort re-evaluation of the original variables in 9,519 consecutive elective surgical patients. *Can J Anaesth*. 2013;60:855-863.
4. Ford MK, Beattie WS and Wijeyesundera DN. Systematic review: prediction of perioperative cardiac complications and mortality by the revised cardiac risk index. *Ann Intern Med*. 2010;152:26-35.
5. Gupta PK, Gupta H, Sundaram A, Kaushik M, Fang X, Miller WJ, Esterbrooks DJ, Hunter CB, Pipinos II, Johanning JM, Lynch TG, Forse RA, Mohiuddin SM and Mooss AN. Development and validation of a risk calculator for prediction of cardiac risk after surgery. *Circulation*. 2011;124:381-387.
6. <https://arxiv.org/abs/2205.03242>
7. <https://github.com/echonet/PreOpNet>

Reviewer #2 (Remarks to the Author):

In their study, Holmstrom et al seek to answer an important medical question – can an easily accessible screen for chronic kidney disease be identified. As the authors accurately note, such a test would be clinically important in enabling early therapy to prevent significant morbidity and mortality. In the present work the investigators apply a convolutional neural network to a standard 12 lead ECG to determine the absence or presence of renal disease in a retrospective cohort, and find the overall test AUC is 0.767, with greater performance in younger cohorts.

The application of convolution neural networks to the 12 lead ECG for several years has been a known powerful screen in retrospective studies for cardiac and non-cardiac disease, and has been demonstrated to apply across diverse, global populations.¹⁻¹⁴ In more recent and novel work, the approach has been shown to identify occult disease in prospective studies, and to correlated to therapeutic drug effects.^{15,16}

The specific question addressed by Holmstrom et al is important and has been recently address by Kwon et al, using similar techniques (retrospective analysis using convolutional neural networks).¹⁷ Interestingly, Kwon et al found an AUC 0.9, substantially higher than the findings of Holmstrom et al. On reason for this might be the much shorter time interval between the ECG and renal assessment (30 minutes for Kwon et al, vs. up to a year in the report by Holmstrom). Holmstrom et al could further clarify their work by addressing the following:

1. How does their work compare to that of Kwon et al, and is the test performance so different?

Reply: Thank you for your excellent suggestions, all of which we have dwelled on and incorporated into the manuscript. Our study complements and reaffirms some of the conclusions from the contemporaneous work (Kwon et al was e-published April 2022 and our preprint was submitted March 2022), in that both we found a signal for CKD in 12-lead ECGs that generalizes across healthcare systems.

Unique features of our study include the validation with larger cohorts (nearly 10x larger external validation cohort) across distinct international sites (USA in our study vs. Korean in Kwon et al) as well as the development of a deep learning algorithm that only incorporates information from the ECG waveform (Kwon et al also introduces age and sex as input variables into the algorithm.) Additionally, as you pointed out, Kwon et al used a different CKD definition (eGFR<45 ml/min/1.73m² within 30 minutes of the ECG) while we used a clinical definition based on physician assessment and billing. This is a potential reason for their better model performance - as an eGFR-only definition could also detect *Acute Kidney Injury* (e.g. due to dehydration, pharmacotherapy, urinary tract obstruction) rather than chronic kidney disease. AKI is also important but we chose to focus on CKD, which might have a more subtle presentation that limits the upper bound of AUC. Together, we think that the findings from these two studies support each other, and we have now expanded our discussion by comparing our results to that of Kwon et al.

Page 10 line 290 (Discussion). *A recent paper by Kwon et al also used data from ECG waveforms in addition to age and sex to develop a DLA to detect changes in eGFR(27), which can include both patients with acute kidney injury (e.g., dehydration, pharmacotherapy, urinary tract obstruction) as well as chronic kidney disease. Their model achieved a slightly higher performance with an AUC of 0.86-0.91, however reaffirms the overall conclusion that renal abnormalities can be detected by CKD within large cohorts across multiple international sites.*

2. Were outpatient and inpatient ECGs used? Inpatient ECGs may have greater dynamic changes due to conditions beyond those being studied, and may be more susceptible to the long interval between ECG and renal assessment. Please comment

Reply: Great question and thank you for the interesting question. Our model was trained from both inpatient and outpatient ECGs, and in this revision, we include analysis of model performance in test populations from both ambulatory and inpatient patients. We show that our model performs well without any significant heterogeneity in performance between

ambulatory patients (AUC of 0.763), or in-hospital patients (AUC of 0.762) (data presented in Supplemental Table 3). We have clarified this aspect in the Methods section.

Page 4 line 116 (Methods): Study cohorts included both ambulatory and in-hospital patients. If a patient had multiple ECGs taken within a year of a CKD diagnosis, each ECG-CKD pair was considered an independent case during model training, but only one was used in the test datasets.

Supplemental Material:

Test task	AUC (95% CI)
eGFR<60 ml/min/1.73 m ² *	0.754 (0.737-0.771)
Albuminuria**	0.734 (0.723 - 0.745)
CKD among ambulatory patients	0.763 (0.74-785)
CKD among in-hospital patients	0.762 (0.752-0.772)
CKD among patients with K >5.5mmol/L***	0.741 (0.698 - 0.787)
CKD among patients with K <=5.5mmol/L***	0.758 (0.747 - 0.768)
White patients	0.764 (0.752 - 0.777)
Black patients	0.760 (0.738 - 0.779)
Asian patients	0.749 (0.715 - 0.782)

Unknown or unspecified race patients	0.769 (0.74 - 0.802)
----------------------

Supplemental Table 3. 12-lead ECG based deep learning model's performance in additional tests in the internal cohort. *within a month of ECG (data available from 23,799 patients), **albumin-to-creatinine ratio >30mg/g (data available from 7,218 patients), ***within a month of ECG. AUC=area under the receiver operating characteristics curve, CI=Confidence interval, CKD=chronic kidney disease, eGFR=estimated glomerular filtration rate.

3. Might other variables in addition to the ECG strengthen model performance?

Reply: An excellent suggestion, based on which we have performed additional experiments. In Kwon et al's work above, it is noted that their model includes age and sex, which might benefit model performance. In this revision, we compare the original model using just ECG waveforms with another model incorporating age, sex, HTN, and diabetes to the model input and report that performance is similar (AUC of 0.79, 95% 0.781-0.798). We postulate this is because deep learning models applied to ECG waveforms have already encoded features of age, sex, hypertension, and diabetes, and integrate this information into the prediction of CKD. This is also replicated in our prior work (Ouyang et al. Arxiv 2022 <https://arxiv.org/abs/2205.03242>) which shows that similar addition of clinical covariates to an ECG waveform model offers a marginal improvement on clinical tasks when the included features are likely also learnable from the ECG waveforms themselves.

Page 7, Line 198 (Results): We also tested the performance of our model separately among diabetic, hypertensive, older patients, who are generally considered as high-risk subgroups. 12-lead based model detected CKD with an AUC of 0.747 (0.707-0.783) among diabetic patients, an AUC of 0.714 (0.701-0.726) among patients with hypertension, and an AUC of 0.706 (0.697-0.716) among patients greater than 60 years old. *When the model was trained with 12-lead ECG waveform, age, sex, diabetes, and hypertension, the model achieved similar discrimination of any stage CKD in the held-out test set with an AUC of 0.79 (0.781-0.798).*

4. Were there differences in performance based on electrolyte abnormalities or mechanism of renal injury? Of race, sex or ethnicity?

Reply: Thank you for this comment. We have provided the model performance based on CKD stage and sex (Table 2, Table 3, Supplemental Table 6, Supplemental Table 7) and based on potassium abnormality, mechanism of kidney injury (albuminuria/reduced eGFR), and race/ethnicity (Supplemental Table 3). The model performed similarly in these subgroups.

Test task	AUC (95% CI)
eGFR<60 ml/min/1.73 m ² *	0.754 (0.737-0.771)
Albuminuria**	0.734 (0.723 - 0.745)
CKD among ambulatory patients	0.763 (0.74-785)
CKD among in-hospital patients	0.762 (0.752-0.772)
CKD among patients with K >5.5mmol/L***	0.741 (0.698 - 0.787)
CKD among patients with K <=5.5mmol/L***	0.758 (0.747 - 0.768)
White patients	0.764 (0.752 - 0.777)
Black patients	0.76 (0.738 - 0.779)
Asian patients	0.749 (0.715 - 0.782)
Unknown or unspecified race patients	0.769 (0.74 - 0.802)

Supplemental Table 3. 12-lead ECG based deep learning model's performance in additional tests in the internal cohort. *within a month of ECG (data available from 23,799 patients), **albumin-to-creatinine ratio >30mg/g (data available from 7,218 patients). ***within a month of ECG. AUC=area under the receiver operating characteristics curve, CI=Confidence interval, CKD=chronic kidney disease, eGFR=estimated glomerular filtration rate.

5. Modeling the use of the test – such as PPV, NPV etc would be useful, as the low PPV (particularly in young cohorts) may limit real-world utilization. I did not see this information in the report

Reply: Thank you for this comment. We now provide AUC, sensitivity, specificity, PPV, and NPV for each tested subgroup in the internal and external test sets (Table 2, Table 3, Supplemental Table 6, Supplemental Table 7). Although the CKD prevalence was <10% in our study cohorts, PPV was relatively good for both the overall cohort (0.443) and young patients (0.57).

Model	AUC (95% CI)	Sensitivity (95% CI)	Specificity (95% CI)	PPV (95% CI)	NPV (95% CI)
12-lead ECG models					
Any stage CKD	0.767 (0.76-0.773)	0.699 (0.699-0.699)	0.698 (0.698-0.698)	0.443 (0.443-0.443)	0.871 (0.871-0.871)
Mild CKD	0.753 (0.735-0.77)	0.75 (0.75-0.75)	0.644 (0.644-0.644)	0.064 (0.064-0.064)	0.987 (0.987-0.987)
Moderate-severe CKD	0.759 (0.75-0.767)	0.785 (0.785-0.785)	0.598 (0.598-0.598)	0.271 (0.271-0.271)	0.936 (0.936-0.936)
ESRD	0.783 (0.773-0.793)	0.704 (0.704-0.704)	0.726 (0.726-0.726)	0.237 (0.237-0.237)	0.953 (0.953-0.953)
High Risk Subgroup analyses for any stage CKD					
Diabetic patients	0.747 (0.707-0.783)	0.699 (0.699-0.699)	0.682 (0.682-0.682)	0.906 (0.906-0.906)	0.342 (0.342-0.342)
Hypertensive patients	0.711 (0.696 - 0.725)	0.648 (0.629 - 0.668)	0.662 (0.644 - 0.68)	0.638 (0.618 - 0.658)	0.672 (0.653 - 0.691)
Age > 60 years	0.706 (0.697-0.716)	0.604 (0.604-0.604)	0.701 (0.701-0.701)	0.397 (0.397-0.397)	0.844 (0.844-0.844)
Male	0.764 (0.755-0.772)	0.727 (0.727-0.727)	0.666 (0.666-0.666)	0.485 (0.485-0.485)	0.849 (0.849-0.849)
Female	0.756 (0.745-0.768)	0.699 (0.699-0.699)	0.680 (0.680-0.680)	0.351 (0.351-0.351)	0.902 (0.902-0.902)
Screening cohort (age < 60)					
Any stage CKD	0.843 (0.836-0.852)	0.761 (0.761-0.761)	0.787 (0.787-0.787)	0.57 (0.57-0.57)	0.899 (0.899-0.899)
Mild CKD	0.795 (0.766-0.823)	0.702 (0.702-0.702)	0.748 (0.748-0.748)	0.075 (0.075-0.075)	0.989 (0.989-0.989)
Moderate-severe CKD	0.854 (0.842-0.865)	0.792 (0.792-0.792)	0.787 (0.787-0.787)	0.373 (0.373-0.373)	0.959 (0.959-0.959)
ESRD	0.842 (0.831-0.853)	0.746 (0.746-0.746)	0.791 (0.791-0.791)	0.394 (0.394-0.394)	0.945 (0.945-0.945)

Table 2. Performance of the 12-lead ECG-based deep learning algorithm in the internal dataset. AUC=area under the receiver operating characteristics curve, CI=Confidence interval, CKD=Chronic kidney disease, ESRD=End-stage renal disease. PPV=positive predictive value. NPV=negative predictive value.

Model	AUC (95% CI)	Sensitivity (95% CI)	Specificity (95% CI)	PPV (95% CI)	NPV (95% CI)
1-lead ECG models					
Any stage CKD	0.744 (0.737-0.751)	0.723 (0.723-0.723)	0.643 (0.643-0.643)	0.41 (0.41-0.41)	0.871 (0.871-0.871)
Mild CKD	0.746 (0.728-0.764)	0.735 (0.735-0.735)	0.66 (0.66-0.66)	0.066 (0.066-0.066)	0.987 (0.987-0.987)
Moderate-severe CKD	0.735 (0.726-0.744)	0.732 (0.732-0.732)	0.618 (0.618-0.618)	0.267 (0.267-0.267)	0.924 (0.924-0.924)
ESRD	0.757 (0.748-0.767)	0.738 (0.738-0.738)	0.647 (0.647-0.647)	0.202 (0.202-0.202)	0.953 (0.953-0.953)
High Risk Subgroup analyses for any stage CKD					
Diabetic patients	0.663 (0.625-0.707)	0.819 (0.819-0.819)	0.457 (0.457-0.457)	0.868 (0.868-0.868)	0.367 (0.367-0.367)
Hypertensive patients	0.684 (0.668 - 0.699)	0.658 (0.639 - 0.677)	0.614 (0.594 - 0.633)	0.61 (0.589 - 0.629)	0.662 (0.641 - 0.68)
Age > 60 years	0.681 (0.671-0.691)	0.679 (0.679-0.679)	0.588 (0.588-0.588)	0.35 (0.35-0.35)	0.849 (0.849-0.849)
Male	0.742 (0.733-0.75)	0.747 (0.747-0.747)	0.612 (0.612-0.612)	0.455 (0.455-0.455)	0.848 (0.848-0.848)
Female	0.735 (0.723-0.747)	0.672 (0.672-0.672)	0.681 (0.681-0.681)	0.342 (0.342-0.342)	0.894 (0.894-0.894)
Screening cohort (age < 60)					
Any stage CKD	0.824 (0.815-0.832)	0.812 (0.812-0.812)	0.705 (0.705-0.705)	0.506 (0.506-0.506)	0.91 (0.91-0.91)
Mild CKD	0.819 (0.791-0.844)	0.771 (0.771-0.771)	0.758 (0.758-0.758)	0.085 (0.085-0.085)	0.991 (0.991-0.991)
Moderate-severe CKD	0.828 (0.816-0.84)	0.814 (0.814-0.814)	0.713 (0.713-0.713)	0.313 (0.313-0.313)	0.96 (0.96-0.96)
ESRD	0.82 (0.808-0.831)	0.82 (0.82-0.82)	0.693 (0.693-0.693)	0.328 (0.328-0.328)	0.955 (0.955-0.955)

Table 3. Performance of the 1-lead ECG-based deep learning algorithm in the internal dataset. AUC=area under the receiver operating characteristics curve, CI=Confidence interval, CKD=Chronic kidney disease, ESRD=End-stage renal disease. PPV=positive predictive value. NPV=negative predictive value.

6. In summary, Halmstrom et al present important confirmatory work of previously published research using well established techniques. As such the work would be meritorious for publication in a nephrology or electrocardiography journal, but lacks the

innovation or novelty in analytic techniques or study design, or prospective clinical application to warrant publication in a high impact journal such as Nature.

Reply: We would respectfully submit that these findings could augment the current approach to CKD screening with the potential to make an impact on clinical care of this important patient population. CKD can indeed be detected in a relatively straightforward manner and blood/urine tests are recommended for high-risk patients. However, only ~20% of such patients receive appropriate testing in the U.S. (PMID 34353883). Therefore, most of the high-risk patients are unaware of their underlying CKD, highlighted by the findings published from two national population-based surveys in the U.S. (BRFSS and NHANES) (PMID 28410862). In addition, a substantial proportion of CKD patients do not belong to the high-risk group and are hence not recommended for screening on a regular basis. Therefore there is a need for novel CKD screening methods.

ECG is an inexpensive, widely available, and noninvasive examination that is frequently used in various clinical circumstances, and it may be used more often than laboratory tests. For example, in our patient cohort from Cedars-Sinai, 74.3% of used ECGs had no serum creatinine or eGFR estimation within 30 days and 50.7% had no serum creatinine or eGFR estimation at any point in the system. We do not suggest that an ECG-based deep learning algorithm could replace blood and urine tests for CKD screening, but rather augment the current CKD screening strategy. Moreover, given the rapid developments in remote ECG recording with wearable devices, we think that ECG-based DL models may as well have the potential to improve prescreening for CKD outside the conventional healthcare unit setting (Supplemental Figure 3). However, further prospective studies are needed to investigate the clinical usefulness of ECG-based DLA in more detail before it can be deployed into clinical practice.

References:

- 1 Ahn, J. C. et al. Development of the AI-Cirrhosis-ECG Score: An Electrocardiogram-Based Deep Learning Model in Cirrhosis. *Am J Gastroenterol* 117, 424-432 (2022). <https://doi.org/10.14309/ajg.0000000000001617>
- 2 Attia, I. Z. et al. External validation of a deep learning electrocardiogram algorithm to detect ventricular dysfunction. *Int J Cardiol* 329, 130-135 (2021). <https://doi.org/10.1016/j.ijcard.2020.12.065>
- 3 Attia, Z. I. et al. Age and Sex Estimation Using Artificial Intelligence From Standard 12-Lead ECGs. *Circ Arrhythm Electrophysiol* 12, e007284 (2019). <https://doi.org/10.1161/CIRCEP.119.007284>
- 4 Attia, Z. I. et al. Screening for cardiac contractile dysfunction using an artificial intelligence-enabled electrocardiogram. *Nat Med* 25, 70-74 (2019). <https://doi.org/10.1038/s41591-018-0240-2>
- 5 Attia, Z. I. et al. An artificial intelligence-enabled ECG algorithm for the identification of patients with atrial fibrillation during sinus rhythm: a retrospective analysis of outcome prediction. *The Lancet* 394, 861-867 (2019). [https://doi.org/10.1016/s0140-6736\(19\)31721-0](https://doi.org/10.1016/s0140-6736(19)31721-0)
- 6 Galloway, C. D. et al. Development and Validation of a Deep-Learning Model to Screen for Hyperkalemia From the Electrocardiogram. *JAMA Cardiol* 4, 428-436 (2019). <https://doi.org/10.1001/jamacardio.2019.0640>

- 7 Cohen-Shelly, M. et al. Electrocardiogram screening for aortic valve stenosis using artificial intelligence. *Eur Heart J* 42, 2885-2896 (2021). <https://doi.org:10.1093/eurheartj/ehab153>
- 8 Attia, Z. I., Harmon, D. M., Behr, E. R. & Friedman, P. A. Application of artificial intelligence to the electrocardiogram. *Eur Heart J* 42, 4717-4730 (2021). <https://doi.org:10.1093/eurheartj/ehab649>
- 9 Adedinsewo, D. et al. Artificial Intelligence-Enabled ECG Algorithm to Identify Patients With Left Ventricular Systolic Dysfunction Presenting to the Emergency Department With Dyspnea. *Circ Arrhythm Electrophysiol* 13, e008437 (2020). <https://doi.org:10.1161/CIRCEP.120.008437>
- 10 Adedinsewo, D. A. et al. Detecting cardiomyopathies in pregnancy and the postpartum period with an electrocardiogram-based deep learning model. *Eur Heart J Digit Health* 2, 586-596 (2021). <https://doi.org:10.1093/ehjdh/ztab078>
- 11 Harmon, D. M. et al. Real-world performance, long-term efficacy, and absence of bias in the artificial intelligence enhanced electrocardiogram to detect left ventricular systolic dysfunction. *European Heart Journal - Digital Health* 3, 238-244 (2022). <https://doi.org:10.1093/ehjdh/ztac028>
- 12 Ribeiro, A. H. et al. Automatic diagnosis of the 12-lead ECG using a deep neural network. *Nat Commun* 11, 1760 (2020). <https://doi.org:10.1038/s41467-020-15432-4>
- 13 Vaid, A. et al. Using Deep-Learning Algorithms to Simultaneously Identify Right and Left Ventricular Dysfunction From the Electrocardiogram. *JACC: Cardiovascular Imaging* 15, 395-410 (2022). <https://doi.org:https://doi.org/10.1016/j.jcmg.2021.08.004>
- 14 Zhu, H. et al. Automatic multilabel electrocardiogram diagnosis of heart rhythm or conduction abnormalities with deep learning: a cohort study. *Lancet Digit Health* 2, e348-e357 (2020). [https://doi.org:10.1016/S2589-7500\(20\)30107-2](https://doi.org:10.1016/S2589-7500(20)30107-2)
- 15 Yao, X. et al. Artificial intelligence-enabled electrocardiograms for identification of patients with low ejection fraction: a pragmatic, randomized clinical trial. *Nat Med* 27, 815-819 (2021). <https://doi.org:10.1038/s41591-021-01335-4>
- 16 Tison Geoffrey, H. et al. Assessment of Disease Status and Treatment Response With Artificial Intelligence-Enhanced Electrocardiography in Obstructive Hypertrophic Cardiomyopathy. *Journal of the American College of Cardiology* 79, 1032-1034 (2022). <https://doi.org:10.1016/j.jacc.2022.01.005>
- 17 Kwon, J. M. et al. Artificial intelligence assessment for early detection and prediction of renal impairment using electrocardiography. *Int Urol Nephrol* 54, 2733-2744 (2022). <https://doi.org:10.1007/s11255-022-03165-w>

Reviewer #3 (Remarks to the Author):

In this paper, the authors assessed the predictive value of machine learning algorithms based on ECG changes to detect chronic kidney disease (CKD). Model performance was reasonable, especially in patients below 60 years of age. Especially as a proof of concept study, the data are interesting

1. Whereas it is quite relevant that ECG changes are associated with even mild CKD, this has been shown in previous studies (e.g. Kestenbaum CJASN 2007) the practical consequence of the study remains somewhat elusive in the paper. This because CKD can be detected in a relatively straightforward and non-expensive manner (blood and urine tests) which is already part of guidelines for high risk groups. Most likely, costs for ECG will be higher, not counting potential additional costs of using a deep learning algorithm, whereas PPV is relatively low . It would be interesting to

address how the use of ECG can overcome current limitations in screening for occult CKD , other than the fact that it may be advisable to measure kidney function in patients with abnormal results on a ECG that was performed for another reason? Do the authors imply to use ECG as a primary screening tool or as an add-on , e.g. in patients who are not under medical care but do have a Smartwatch with lead I ECG available?

Reply: Thank you for your thoughtful and constructive review. These are excellent suggestions, all of which are worthy of discussion and incorporated into the manuscript. CKD can indeed be detected in a relatively straightforward manner and blood/urine tests are recommended for high-risk patients. However, although screening high-risk patients is recommended in the current guidelines, only ~20% of such patients receive appropriate testing in the U.S. (PMID 34353883). Therefore, most of the high-risk patients are unaware of their underlying CKD, and this was also observed in two national population-based surveys in the U.S. (BRFSS and NHANES) (PMID 28410862). In addition, a substantial proportion of CKD patients do not belong to the high-risk group and are hence not recommended for screening on a regular basis. Therefore there is a need for discovery of novel CKD screening methods complementary to standard laboratory testing.

ECG is an inexpensive, widely available, and noninvasive examination that is frequently used in various clinical circumstances, and it may be used more often than laboratory tests. For example, in our patient cohort from Cedars-Sinai, 74.3% of used ECGs had no serum creatinine or eGFR estimation within 30 days and 50.7% had no serum creatinine or eGFR estimation at any point in the system. We do not suggest that an ECG-based deep learning algorithm could replace blood and urine tests for CKD screening, but rather augment the current CKD screening strategy. Moreover, given the rapid developments in remote ECG recording with wearable devices, we think that ECG-based DL models may as well have the potential to improve pre-screening for CKD outside the conventional healthcare unit setting (Supplemental Figure 3). However, further prospective studies are needed to investigate the clinical usefulness of ECG-based DLA in more detail before it can be deployed into clinical practice.

In summary, we agree with this reviewer and have now revised our conclusions for clarity. We have also discussed the limitations of the current screening strategy to highlight the importance of novel tools that could improve early CKD diagnosis.

Page 2, Line 54 (Abstract, Conclusions): *Our deep learning algorithm was able to detect CKD using ECG waveforms, with stronger performance in younger patients and more severe CKD stages. This ECG algorithm has the potential to augment screening for CKD.*

Page 9, Line 268 (Discussion): *Low awareness of CKD and limitations in current screening measures highlight the urgency of novel screening strategies to increase detection rates of early-stage CKD. Being non-invasive and often obtained in the clinic, ECGs are often the first line of clinical evaluation. In our healthcare system, 74% of ECGs obtained did not have laboratory testing of kidney function within 30 days. Previous studies have demonstrated that the cost-effectiveness of CKD screening is highly dependent on patient risk factor profile and CKD probability, and there has been debate on whether CKD screening should be targeted only to high-risk patients, or also extend to patients without risk factors for CKD(22-25). Although screening high-risk patients is guideline-recommended, testing rates remain low as only about 20% of high-risk patients receive guideline-recommended assessment in the U.S.(26). Consequently, most of the high-risk patients are likely to be unaware of underlying*

CKD(2, 3). Moreover, a substantial proportion of all CKD patients are not high-risk patients and hence not recommended to be screened regularly, which further highlights the need for novel screening methods.

Page 11, Line 335 (Discussion, Conclusions): Our ECG-based deep learning model was able to detect CKD with good discrimination accuracy in multiple study populations and with particularly high accuracy in patients under 60 years of age. These results suggest that deep learning-based ECG analysis may provide additional value in detecting various CKD stages, especially in younger patients. *The clinical significance of this study lies in the potential enhancement of screening methods for the early detection of CKD, which is crucial to enable early treatment and prevent disease progression.*

2. In the primary cohort characteristics (p6), 17860 patients were diagnosed with CKD, but 51% of (patients with?) ECGs did not have any measurement of renal function in the system. How was the diagnosis of CKD confirmed in these patients. Were those patients (diagnosis CKD, but no data on renal function) included in the analysis?

Reply: An excellent point. Study subjects were determined to have CKD if they had clinician inputted diagnosis in the electronic health record within 1 year of ECG recording. This diagnostic depends on expert adjudication (as oftentimes paper records, OSH records, and additional information is not readily available in the EHR. 51% refers to the proportion of all ECGs (with or without associated CKD diagnosis) that did not have any measurement of renal function in the system, however given the inclusion criteria for the study, had an ICD9/10 code description of degree of kidney dysfunction. We have now revised it to provide these clarifications.

Page 6, Line 165 (Results): In total, we identified 17,860 patients with a CKD diagnosis at Cedars-Sinai Medical Center (7.8% of the total patient sample), among which 7,816 had an ECG performed within a 1-year window of CKD diagnosis. Our primary cohort consisted of a total of 247,655 ECGs, of which 221,974 were randomized to the training set (for both training and validation) and 25,681 to the testing set. *Of the primary cohort ECGs, 74.3% had no serum creatinine or eGFR estimation within 30 days and 50.7% of ECGs had no serum creatinine or eGFR estimation at any point in the EHR, however this does not capture outside hospital or paper clinic records of laboratory testing that might have been used in the diagnosis of CKD. The mean age of the primary cohort was 61.3 ±19.7 years and 48% were female. Demographic and clinical characteristics are presented in Table 1. Demographics and clinical characteristics according to age group are presented in Supplemental material (Table 2).*

3. In the same paragraph, it is mentioned that 50.7% of ECGs had no serum creatinine or GFR estimations, which should be rephrased. Does this mean "50.7 of patients with ECGs" or is the patient number lower given the fact that patients might have undergone multiple ECG measurements?

Reply: Thank you for this important point. These metrics are on the ECG level rather than patient level, since a patient might have a laboratory testing outside of the time window of

linking diagnosis to ECG. Our clinical disease definition depends on clinician assessment, and eGFRs are an important but incomplete metric as it does not capture outside healthcare system measurements. This 50.7% refers to the proportion of ECGs in the internal study cohort (with or without CKD diagnosis), that had no associated serum creatinine or eGFR estimation at any point in the system. We have now revised this sentence in the Results section (see above).

Page 6, Line 165 (Results): *In total, we identified 17,860 patients with a CKD diagnosis at Cedars-Sinai Medical Center (7.8% of the total patient sample), among which 7,816 had an ECG performed within a 1-year window of CKD diagnosis. Our primary cohort consisted of a total of 247,655 ECGs, of which 221,974 were randomized to the training set (for both training and validation) and 25,681 to the testing set. Of the primary cohort ECGs, 74.3% had no serum creatinine or eGFR estimation within 30 days and 50.7% of ECGs had no serum creatinine or eGFR estimation at any point in the EHR, however this does not capture outside hospital or paper clinic records of laboratory testing that might have been used in the diagnosis of CKD. The mean age of the primary cohort was 61.3 ± 19.7 years and 48% were female. Demographic and clinical characteristics are presented in Table 1. Demographics and clinical characteristics according to age group are presented in Supplemental material (Table 2).*

4. Were patients who did not have any information on eGFR but underwent ECG included in the analysis? If so, would it not be better to include only patients in whom renal function measurements (either indicative of CKD or not) in the analysis? This because one cannot be sure that the patients without CKD diagnosis or ECG measurements indeed have a normal renal function, given the high prevalence of CKD in the general population?

Reply: An excellent point and we agree. Yes, patients without information on eGFR were included in the main analysis, with the majority of those patients as control patients. Given that a substantial proportion of non-CKD patients with an ECG do not have an eGFR measurement, excluding those patients may skew the dataset, however is most reflective of a general screening population. To answer this important point, we include an additional analysis that includes only patients with eGFR measurement (including normal eGFR controls) within a month of ECG in Supplement Table 3. The model had a similar performance in identifying patients with eGFR <60 ml/min/1.73 m² from patients with normal eGFR (AUC of 0.754).

Supplement Table 3.

Test task	AUC (95% CI)
-----------	--------------

eGFR<60 ml/min/1.73 m ² *	0.754 (0.737-0.771)
Albuminuria**	0.734 (0.723 - 0.745)
CKD among ambulatory patients	0.763 (0.74-785)
CKD among in-hospital patients	0.762 (0.752-0.772)
CKD among patients with K >5.5mmol/L***	0.741 (0.698 - 0.787)
CKD among patients with K ≤5.5mmol/L***	0.758 (0.747 - 0.768)
White patients	0.764 (0.752 - 0.777)
Black patients	0.76 (0.738 - 0.779)
Asian patients	0.749 (0.715 - 0.782)
Unknown or unspecified race patients	0.769 (0.74 - 0.802)

Supplemental Table 3. 12-lead ECG based deep learning model's performance in additional tests in the internal cohort. *within a month of ECG (data available from 23,799 patients), **albumin-to-creatinine ratio >30mg/g (data available from 7,218 patients). ***within a month of ECG. AUC=area under the receiver operating characteristics curve, CI=Confidence interval, CKD=chronic kidney disease, eGFR=estimated glomerular filtration rate.

5. In table 1, the classification of CKD (moderate/end stage) i.e. 6952 patients does not seem to match with eGFR criteria where 9010 patients had a eGFR <60 ml/min/1.73m². The prevalence of ESRD appears to be very high (2.7% in a population study!) as compared to the other stages. In the table, stage 5 CKD is included in the "moderate" category, which is incorrect. The fact that a different type of percentages are mentioned

for the eGFR (adding up to 99,2%) as compared to the stages of CKD (where they indicate percentages of the population) is somewhat confusing

Reply: Thank you for this important clarification. Much of the variation between classification likely results from ECG-level vs. patient level analysis and the incidence of acute kidney injury that are included to increase the number patients with at least one eGFR<60 ml/min/1.73m² but however never met the definition of chronic moderate-severe/end-stage kidney diseases. A single eGFR measurement is not sufficient for a CKD diagnosis and some of the patients with a single eGFR measurement of <60 do not meet the criteria for CKD (eGFR<60 or albuminuria for at least 3 months). Especially in acute clinical conditions, eGFR may be temporarily reduced due to other factors than CKD (e.g. dehydration, urinary tract obstruction, or pharmacotherapy).

The relatively high prevalence of ESRD may be explainable by the fact that our study cohort included only patients with an ECG recording. Patients with ESRD are more likely to have an ECG recording than healthy patients, which leads to a higher prevalence than in the general population. We also demonstrated in an additional analysis that the model performance was not dependent on the CKD prevalence and remained similar across the variable CKD proportions (Supplement Table 4).

We agree that stage 5 CKD is not moderate CKD. This is our error, thank you for catching it. Instead of “moderate CKD”, stage 3-5 CKD should be labeled as “moderate-severe CKD” in Table 1. We have now revised this aspect in the manuscript.

Supplemental material:

Proportion of available negative patients	CKD prevalence	AUC (95% CI)
100%	6.0%	0.770 (0.764-0.776)
50%	11.4%	0.769 (0.763-0.774)
25%	20.5%	0.769 (0.763-0.774)
10%	39.1%	0.764 (0.758-0.770)

Supplemental Table 4. Performance of different ECG-based deep learning models to detect any stage CKD. Models were trained with varying levels of CKD prevalence to illustrate the effect of CKD prevalence in the training set on the model performance. AUC=area under the receiver operating characteristics curve, CI=Confidence interval, CKD=chronic kidney disease.

REVIEWERS' COMMENTS:

Reviewer #1 (Remarks to the Author):

None

Reviewer #3 (Remarks to the Author):

Thanks for your consideration of my comments. I have no further questions, but suggest to give more emphasis to supplementary table 3 in the revised paper, as this appears to be a vital addition in my view . A CKD diagnosis based on ICD 10 without any data on renal function or albuminuria is, at least in the eyes of the reviewer, still somewhat hazardous.

REVIEWERS' COMMENTS:

Reviewer #1 (Remarks to the Author):

None

Reply: Thank you for your thoughtful review

Reviewer #3 (Remarks to the Author):

Thanks for your consideration of my comments. I have no further questions, but suggest to give more emphasis to supplementary table 3 in the revised paper, as this appears to be a vital addition in my view . A CKD diagnosis based on ICD 10 without any data on renal function or albuminuria is, at least in the eyes of the reviewer, still somewhat hazardous.

Reply: Thank you for your thoughtful review. We have now given more emphasis to the results in the supplementary table 3 and discussed these in the Results section. We have now also highlighted the limitations related to the CKD definition and discussed this aspect in more detail in the limitations section.

***Page 7 line 216 (Results):** The model performed similarly in detecting CKD in subset populations of patients with albuminuria, patients with corresponding laboratory testing and documented eGFR, and in both ambulatory and in-hospital patients (Supplementary Table 3). In patients with both a CKD diagnosis and eGFR estimated to be less than 60mL/min, the AUC was 0.754 (0.737 – 0.771), and this performance was similar in patients with hyperkalemia with an AUC of 0.741 (0.698 – 0.787) and without hyperkalemia with an AUC of 0.758 (0.747 – 0.777). The model also performed well in patients with known albuminuria, with an AUC of 0.734 (0.723 – 0.745) and had similar performance regardless of the positive to negative ratio in the training set (Supplementary Table 4).*

***Page 11 line 317 (Limitations):** However, a few limitations warrant consideration. Our study is retrospective, and study populations are derived from two large academic medical centers situated in dense urban metropolitan areas using ICD-9 codes. By prioritizing priority codes, we sought to avoid incidences of acute rather than chronic kidney injury, however, we cannot exclude the possibility that some of the study subjects without CKD diagnosis in electronic health records have an undiagnosed disease, as especially mild-stage CKD can often be undiagnosed, particularly using an ICD9 code-based adjudication. In the subset with both ICD-9 code adjudication of CKD as well as laboratory testing, the ICD-9 codes were consistent with the calculated eGFR, however, only a minority of patients were able to be linked to data regarding microalbuminuria.*